# Evolution of Antibiotic Resistance and the Relationship between the Antibiotic Resistance Genes and Microbial Compositions under Long-Term Exposure to Tetracycline and Sulfamethoxazole

**DOI:** 10.3390/ijerph16234681

**Published:** 2019-11-25

**Authors:** Bingbing Du, Qingxiang Yang, Ruifei Wang, Ruimin Wang, Qiang Wang, Yuan Xin

**Affiliations:** 1School of Environment, Henan Normal University, Xinxiang 453007, China; db85250820@126.com; 2College of Life Sciences, Henan Normal University, Xinxiang 453007, China; wswrf@163.com (R.W.); wangruimin@htu.edu.cn (R.W.); wangqiang@htu.edu.cn (Q.W.); joy940671775@163.com (Y.X.); 3College of Basic Medical Science, Luohe Medical College, Luohe 462002, China

**Keywords:** anoxic-aerobic systems, antibiotic pressure, antibiotic resistance gene, network analysis

## Abstract

The removal of antibiotics and widespread of antibiotic resistance genes (ARGs) have received continuous attention due to the possible threats to environment. However, little information is available on the evolution of antibiotic resistance and the relationship between ARGs and microbial communities under long-term exposure to sub-inhibitory concentrations of antibiotics. In our study, two laboratory-scale anoxic-aerobic wastewater treatment systems were established and operated for 420 days to investigate the evolution of antibiotic resistance under exposure of 5 mg·L^−1^ tetracycline (TC) or 5 mg·L^−1^ TC and 1 mg·L^−1^ sulfamethoxazole (SMX). The average removal rates of TC and SMX were about 59% and 72%, respectively. The abundance of the main ARGs responsible for resistance to TC and SMX increased obviously after antibiotics addition, especially when TC and SMX in combination (increased 3.20-fold). The *tetC* and *sul1* genes were the predominant genes in the development of TC and SMX resistance, in which gene *sul1* had the highest abundance among all the detected ARGs. Network analysis revealed that under antibiotic pressure, the core bacterial groups carrying multiple ARGs formed and concentrated in about 20 genera such as *Dechloromonas*, *Candidatus Accumulibacter*, *Aeromonas*, *Rubrivivax*, in which *intI1* played important roles in transferring various ARGs except *sul3*.

## 1. Introduction

Antibiotics are widely applied to treat infections or employed as prophylactics in human and animal medicine, among which tetracycline (TC) and sulfamethoxazole (SMX) are used most frequently [1,2,3,4,5,6]. However, antibiotics are often misused or overused and the majority cannot be metabolized effectively, resulting in high concentrations of residual antibiotics in the environment [7,8,9]. There is an indisputable fact that the abuse of antibiotics can lead to proliferation, accumulation and widespread distribution of antibiotic resistance bacteria and antibiotic resistance genes (ARGs), which is becoming an increasing threat to global health and posing a serious threat to the environment [10,11,12,13,14]. The traditional treatment processes used in wastewater treatment systems are regarded as the important ways for antibiotic pollutants to enter the environment, and they were also well known as important reservoirs for all kinds of antibiotic contaminants, antibiotic resistant bacteria and the corresponding ARGs [15,16].

In fact, the occurrence of antibiotics inevitably affects the functional microorganisms in activated sludge and contributes to the proliferation and dissemination of ARGs in wastewater treatment systems [10,12,17,18]. The transmission of ARGs is achieved mainly through vertical gene transfer and horizontal gene transfer, in which horizontal gene transfer is the predominant mechanism for transmission of ARGs among different genera by facilitating the exchange of mobile genetic elements (including *intI1*, *ISCR1* and *Tn916*/*1545*) [4,19]. Furthermore, researchers commonly believe that the proliferation of ARGs can be induced by antibiotics even at sub-inhibitory concentrations and may still exist in the environment even though the pressure of antibiotics has been removed [9,20,21,22].

With the increasing attention on antibiotics and ARGs pollution, many studies have been found to research the impact of antibiotics on the performance of activated sludge systems, as well as the existence of ARGs in activated sludge under antibiotic exposure [17,23,24,25,26,27]. Unfortunately, only approximately 1% of bacteria could be cultured in the laboratory and this limitation has restricted the understanding about the detailed metabolism and functions of those microorganisms [28,29]. In particular, the evolution of antibiotic resistance and the relationship between ARGs and microbial community compositions in activated sludge are not clear, especially under long-term antibiotic pressure.

Adding a single antibiotic is beneficial for researching under controlled conditions. TC was preferentially selected as the representative antibiotic because this broad-spectrum antibiotic was the most widely used and frequently detected antibiotic around the world [6,23,30]. However, exposure to a combination of antibiotics is more representative of practical cases. Great curiosity was aroused about the difference between the impacts of a single antibiotic and combined antibiotics on the wastewater treatment system. SMX was then chosen as another representative antibiotic for the season that it was the most widely found sulfonamides in wastewater treatment and has ecological toxicity to the environment [3,12]. Previous studies have investigated the effects of sub-inhibition dosages of TC and SMX exposure on the performances and activated sludge characteristics in a laboratory-scale anoxic-aerobic (A-O) wastewater treatment system [25,31]. The results indicated that although there were no significant effects on the main pollutant removal (such as chemical oxygen demand and total nitrogen), long-term exposure to TC alone or to TC combined with SMX did affect the sludge settling performance by affecting the compositions of microbial communities. However, it was not clear about the evolution of antibiotic resistance and the relationship between ARGs and microbial community compositions in activated sludge, especially under long-term antibiotic pressure. Tracking the shifts of key microbial genera and further investigating the succession of the potential ARGs hosts will help us to understand the evolution process and the mechanism of ARGs transmission under antibiotic pressure in wastewater treatment system. Therefore, combined antibiotics TC and SMX were subsequently added into the same laboratory-scale A-O wastewater treatment system for further research.

The objectives of this study were to further investigate the removal of TC and SMX, the evolution of antibiotic resistance and the relationship between the main ARGs and the microbial compositions in the above A-O system. High-throughput genetic sequencing, quantitative polymerase chain reaction (qPCR) analysis, and network analysis were applied to explore the host relationship with ARGs and help clarify the mechanism of ARGs transmission between microbial communities under antibiotics selection pressure. Our report is the first comparative study about the evolution of antibiotic resistance under long-term exposure of microorganisms to sub-inhibition concentrations of a single antibiotic (TC) and combined antibiotics (TC and SMX). The results of this study will provide improved treatment guidance to reduce environmental risk for wastewater treatment plants that receive wastewater containing antibiotic residues, such as wastewater arising from hospitals and livestock farming.

### Highlights

The evolution of antibiotic resistance was studied in an anoxic-aerobic system.The average removal rates of TC and SMX were about 59% and 72%, respectively.The ARGs abundance for TC and SMX increased significantly with antibiotic exposure.The ARGs were possibly related to 20 main bacterial genera under antibiotic pressure.Gene *intI1* played important roles in horizontal transfer of various ARGs except *sul3*.

## 2. Materials and Methods

### 2.1. Design of the Experimental Approach

Experiments were conducted in two identical laboratory-scale A-O systems (System A and System B) as shown schematically in Figure 1. The detailed information was described previously [25,31]. System A was set as the control, while system B was fed with 5 mg·L^−1^ TC from day 41 to 200 and a combination of 5 mg·L^−1^ TC and 1 mg·L^−1^ SMX from day 241 to 420. Recovery period was from day 201 to 240, in which 50% of the activated sludge in both systems was exchanged with fresh sludge in order to keep the sludge states similar as much as possible. Both systems were operated at room temperature (about 25 °C) in a darkened room that excluded light and prevented photolysis of TC and SMX. The whole operational period (420 days) was divided into four phases: Phase 1 (no antibiotic addition, days 0–40), phase 2 (addition of 5 mg·L^−1^ TC in system B, days 41–200), phase 3 (recovery phase, no antibiotic addition, days 201–240), and phase 4 (addition of 5 mg·L^−1^ TC and 1 mg·L^−1^ SMX in system B, days 241–420).

The intention of the study was to investigate comparatively the evolution of antibiotic resistance associated with their corresponding resistance genes when exposed to a single antibiotic (TC) and a combination of antibiotics (TC and SMX) in a lab-scale A-O wastewater treatment system. Considering that the difference of antibiotics concentration in actual wastewater treatment systems especially in hospital and livestock wastewaters, the selective antibiotics concentrations in our experiment were estimated from published reports [13,23,32,33,34].

### 2.2. Total DNA Extraction and Bacterial Compositions Analysis

Sludge samples were taken on days 1, 100, 200, 240, 330, and 420 of the system operation time. The total DNA was extracted using the E.Z.N.A.^®^ Soil DNA Kit (Omega Bio-tek, Inc., Norcross, GA, USA). The purity and yield of the DNA were determined by spectrophotometric analysis (Nano Drop 2000, Thermo Fisher Scientific, MA, USA). According to the ratio of light absorbance at 260 (A260) and 280 nm (A280), the purity of nucleic acid was estimated, and the ratio of pure DNA was between 1.8 and 2.0. The extracted DNA was detected using 1% (w/v) agarose gel electrophoresis at 110 V for 40 min and the purified DNA samples were immediately stored at −20 °C until the extracts were sent for high-throughput sequencing.

### 2.3. Concentration Determination of Tetracycline and Sulfamethoxazole

Solid phase extraction and high performance liquid chromatography (HPLC) were used to extract and determine the concentration of TC and SMX in water samples according to the procedure of Wang et al. (2017) [35]. TC and SMX were purchased from Dr. Ehrenstorfer (Augsburg, Germany) as standard compounds, and were dissolved to 100 mg·L^−1^ using methanol in a brown flask as the stock solution, which was stored at −20 °C. The calibration standards of TC (0.0, 0.2, 0.5, 1.0, 2.0, and 5.0 mg·L^−1^) and SMX (0.0, 0.2, 0.5, 1.0, 2.0, and 2.5 mg·L^−1^) were prepared by a serial dilution of the stock solution. The analysis of TC and SMX was performed on an Agilent 1260 (Agilent Technologies, Santa Clara, CA, USA) HPLC system. Agilent ChemStation software integral to the HPLC was used to acquire and process data. The analytical procedures for TC and SMX were performed as previously described [36,37,38], and detailed information about the analyses is shown in Appendix A.

### 2.4. Quantification of Antibiotic Resistance Genes

The abundances of genes were analyzed by qPCR, including six TC resistance genes (*tetA*, *tetC*, *tetL*, *tetO*, *tetW*, *tetX*), three sulfonamide resistance genes (*sul1*, *sul2*, *sul3*), and the integrase gene of class 1 integrons (*intI1*). Primer sequences targeting these genes are listed in Table 1 and these were synthesized by Sangon Biotech Co., Ltd. (Shanghai, China). The qPCR reaction was performed on the LightCycler^®^ 96 qPCR System (F. Hoffmann-La Roche AG, Basel, Switzerland) with SYBR^®^ Green PCR master mix (Vazyme, Nanjing, China) following the protocol reported previously [31,39]. In brief, the qPCR reaction mixture (20.0 μL) contained 10.0 μL SYBR^®^ Green Master Mix, 0.4 μL forward/reverse primer (10 μM), 2.0 μL template DNA, and 7.2 μL double distilled water. The procedure for qPCR amplification was set according to the kit and the thermo cycling steps were as follows: ① Preincubation: 95 °C for 300 s; ② two-step amplification for 45 cycles: 95 °C for 10 s and 60 °C for 30 s; ③ melting: 95 °C for 15 s, 60 °C for 60 s, and 95 °C for 15 s; and ④ cooling: 37 °C for 30 s. All the processing steps were carried out while the samples were on ice and all samples including the negative controls were quantified in triplicate to guarantee low technical variance.

### 2.5. Statistical Analysis

The figures were plotted using Origin Pro 9.1 software (Origin Lab Corp., Northampton, MA, USA). To investigate the relationship among microbial community members and ARGs, a correlation matrix was constructed by calculating all possible pair-wise Spearman’s rank correlations using SPSS 20.0 statistical software (IBM Corp., Armonk, NY, USA). A *p* value < 0.05 was considered statistically significant. Network analysis based on the Spearman analysis was performed using Gephi V0.9.2 open source software (https://gephi.org/).

## 3. Results and Discussion

### 3.1. Efficiency of Antibiotics Removal in the Anoxic-Aerobic Wastewater Treatment System

The target TC and SMX concentration in the influent and effluent were measured every 15 days from day 41 to 420 to determine the antibiotics removal capacity of system B (Figure 2). The average TC concentrations were 24.35 and 10.19 μg·L^−1^ in the influent and effluent, respectively, with the removal rate of 58.40% when exposed to TC alone from day 41 to 200. Under exposure to combined TC and SMX (days 241–420), the average TC removal rate was 59.11% with some fluctuations, suggesting that TC removal was similar during these two conditions. However, the mean removal rate of SMX (72.40%, ranging from 63.66% to 82.19%) was higher than that of TC under the combined exposure of TC and SMX, resulting in an average effluent SMX concentration of 1.07 μg·L^−1^ in system B.

Previous reports have indicated that the antibiotic removal mainly depends on biodegradation and adsorption in wastewater treatment systems [41,42]. The adsorption process usually occurs rapidly after antibiotics addition, and the biodegradation of antibiotics occurs gradually and continuously. Sulfonamides were reported to be mainly removed by biodegradation and tetracyclines by adsorption [18,41]. The negligible adsorption of sulfonamides to sludge may be associated with its high solubility in water; in contrast, the significant rapid adsorption of tetracycline onto activated sludge is due to its molecular form and complexation [43].

In this study, the similarity of mean TC removal by microorganisms in both conditions (exposed to TC alone and the combination of TC and SMX) revealed the stable sorption performance of activated sludge under long-term antibiotic pressure. The SMX removal rate was 23.23% higher than that of TC. This was probably because the functional amino group on the aromatic ring of SMX could be oxidized by autotrophic nitrifying bacteria, and some microbial communities probably utilized SMX as energy or carbon sources for growth [7,44].

### 3.2. Evolution of Antibiotic Resistance Genes under Long-Term Exposure to Antibiotics in the Anoxic-Aerobic Wastewater Treatment System

To estimate the evolution of ARGs under antibiotic pressure, the relative abundances of six TC resistance genes (*tetA*, *tetC*, *tetL*, *tetO*, *tetW,* and *tetX*), three sulfonamide resistance genes (*sul1*, *sul2,* and *sul3*), and the integrase gene of class 1 integron (*intI1*) were determined using qPCR analysis in both systems. The relative abundances (gene copies of ARGs normalized to the gene copies of 16S rRNA) of ARGs in different phases for system A and system B are shown in Figure 3 and Appendix A. Notably, the abundance of the detected ARGs responsible for TC and SMX resistance increased obviously in the presence of added antibiotics especially when TC and SMX were added simultaneously, demonstrating that the proliferation and dissemination of ARGs were accelerated strongly under antibiotic selection pressure.

According to the different functional mechanisms, the detected TC resistance genes were divided into three groups that included three efflux pump genes (*tetA*, *tetC,* and *tetL*), two ribosomal protection protein genes (*tetO* and *tetW*), and a single enzymatic inactivation gene (*tetX*) [45,46]. The average abundances of these TC resistance genes in both system A and system B were in the order of *tetC* > *tetA* > *tetX* > *tetL* > *tetW* > *tetO*. The most obvious increase occurred in the relative abundance of *tetC* (*p* < 0.05), which increased 8.64-fold from an average of 4.28 × 10^−3^ copies/16S rRNA (system A) to 3.69 × 10^−2^ copies/16S rRNA (system B) under antibiotic pressure (Figure 3 and Appendix A). Meanwhile, compared to the abundance under exposure to only TC, a 2.82-fold increase for *tetC* was observed under the combined exposure in system B. The next largest change in abundance was that of TC resistance gene *tetA*, which exhibited a 1.97-fold increase under exposure to antibiotics in system B compared to the abundance in system A. The *tetA* and *tetC* genes both belonged to the group of efflux pump genes; these were regarded as the predominant TC resistance genes and spread earlier and quicker to encode resistance to TC [4,47]. Previous reports also revealed that the enrichment of *tetA* and *tetC* genes was enhanced by the addition of antibiotics [46,48].

The *tetX* gene is a unique enzymatic inactivation gene that encodes tetracycline-degrading monooxygenase and can catalyze the inactivation of varieties of tetracyclines [49]. With the antibiotic additions in system B, the average relative abundance of *tetX* was 2.93 × 10^−3^ copies/16S rRNA, which was 1.51 times larger (*p* > 0.05) than the abundance in control system A (1.94 × 10^−3^ copies/16S rRNA). Moreover, the average abundance of other TC genes (*tetL*, *tetO,* and *tetW*) was also higher in system B than in system A during the whole experimental period except for some individual phase. For example, compared to that in system A, the abundance of *tetL* decreased by 35.87% under exposure to TC alone in phase 2 (days 41–200). Certainly, under the pressure of long-term antibiotics exposure, the proliferation of TC resistance genes increased obviously. This demonstrated that antibiotics addition might promote positively the accumulation and transmission of TC resistance genes during the whole experimental period in wastewater treatment system.

The abundances of three sulfonamide resistance genes (*sul1*, *sul2,* and *sul3*) all significantly (*p* < 0.05) increased as the system operation time increased, implying an enrichment of sulfonamide resistance genes in system B when exposed to antibiotics. The trend of relative abundances for the three sulfonamide resistance genes followed the order *sul1* > *sul2* > *sul3*, which was consistent with previous studies that showed *sul1* was the most prevalent gene among sulfonamide resistance genes [12,39]. It is important to note that *sul1* had the highest average abundance among all the detected ARGs throughout the whole 420-day experiment even in the control system A where no antibiotic was added (Figure 3). The *sul1* gene is a plasmid-borne gene that encodes alternative drug-resistant variants of the dihydropteroate synthase enzymes and is accompanied by the presence of integrons that play an important role in the wide distribution of antibiotic resistance [4,9].

Moreover, exposure to the combined TC and SMX caused distinct increases of the three sulfonamide resistance genes as indicated by increases in the average abundance of the *sul1* (2.58-fold), *sul2* (3.17-fold), and *sul3* (1.44-fold) genes, compared to the abundancies when exposed to TC alone in system B. This implied that the sulfonamide resistance genes were mainly driven by the presence of SMX in the wastewater treatment system. A similar result showed that over the course of one year, the concentrations of ARGs encoding tetracycline, sulfamethoxazole, and erythromycin increased, and that the triple effects of antibiotics on proliferation of the ARGs were much greater than the dual effects [15]. It has been universally acknowledged that the transfer and transmission of ARGs can promote multiple mechanisms of antibiotic resistance among microorganisms rapidly and widely though mobile genetic elements [13].

The existence of *intI1* (one of the most common types of mobile genetic elements for ARG capture) could be considered as playing an important role in the widespread proliferation of antibiotic resistance though horizontal gene transfer. There was a slight increase of 1.18-fold (compared to the abundance in system A) for the average abundance of *intI1* in system B when exposed to antibiotics. This increase was mainly due to the exposure of TC and SMX in combination, which increased the abundance of *intI1* to 2.50 times than in system A. Exposure to TC alone dramatically decreased the abundance of *intI1* by 76.35% in the system from day 41 to 200 (Figure 3). The evolution of *intI1* under antibiotic pressure was inconsistent with previous reports. Zhu et al. (2018) reported that the abundance of *intI1* increased by 0.6 orders of magnitude after SMX and TC were added [13]. Similar enrichment of *intI1* was reported under pressure of TC [46], the combination of tetracycline, sulfanilamide, and cefotaxime [9], and the combination of sulfamethoxazole, tetracycline, and erythromycin [15]. However, the relative abundance of *intI1* significantly decreased by 1.005 log in reactors which tetracyclines and sulfonamides were added [16].

Results from the present study indicated that exposure to a single antibiotic (TC) had only a limited effect on the spread of *intI1*, while the combined exposure to TC and SMX stimulated the proliferation of *intI1* through horizontal gene transfer. In particular, *intI1* exhibited significant correlation with *sul1* and *sul2*, especially for *sul1*, which may serve as a partial structure in the 3′-conserved segment region of the class 1 integron [15,50]. This indicated that the addition of SMX accelerated the propagation of sulfonamide resistance genes, which could facilitate the enrichment of *intI1*.

The total abundance of ARGs was also estimated at different phases to further elucidate the evolution of antibiotics resistance. During the initial phase (without antibiotic addition, days 0–40), the abundances of ARGs were similar in both systems and *sul1* was the most abundant gene with average abundances of 1.42 × 10^−2^ copies/16S rRNA (system A) and 1.92 × 10^−2^ copies/16S rRNA (system B). In phase 2 (days 41–200), the total abundance of ARGs increased about 2.02-fold compared to that in system A. One of the tetracycline efflux pump genes *tetA* increased remarkably by 4.28-fold from 6.15 × 10^−3^ to 2.63 × 10^−2^ copies/16S rRNA in system B. However, the abundances of other tetracycline ARGs were stable or even decreased when microbes were exposed to TC alone, indicating that *tetA* was the predominant gene in the development of TC resistance. It is noteworthy that the *sul1* gene was detected as the dominant ARG in both system A and system B even though there was no sulfonamides exposure during this phase; therefore, *sul1* had already been widely distributed in the activated sludge.

During the recovery phase (days 201–240), the abundances of the detected ARGs substantially decreased in system A, while the propagation of most ARGs (especially *tetA* and *sul2*) in system B exhibited no such decrease or delay, even though the pressure of antibiotics had been removed for more than one month. The combined exposure to TC and SMX in system B during phase 4 (days 241–420) greatly stimulated propagation and transmission of ARGs with a 3.20-fold increase, compared to the abundances in system A, in which relatively high abundances of *tetC* and *sul1* were also observed. These results suggested that compared to ribosomal protection protein genes (*tetO* and *tetW*) or the enzymatic inactivation gene (*tetX*) in this study, the efflux pump genes (*tetA* and *tetC*) were more active in the propagation and transmission for TC resistance genes under antibiotic pressure. For sulfonamide resistance, *sul1* was consistently a predominant gene in the presence or absence of antibiotic pressure, while *sul2* was obviously induced by the exposure of SMX. These differences should receive more attention, especially during treatment of livestock farming wastewater and hospital wastewater containing high concentrations of antibiotics, because the transferred resistance might pose substantial risks to the environment.

### 3.3. The Co-Occurrence of Microbial Communities and Their Corresponding Antibiotic Resistance Genes under the Pressure of Tetracycline and Sulfamethoxazole in the Anoxic-Aerobic Wastewater Treatment System

The relative abundances of the dominant genera were listed in order to understand the evaluation of microbial community structure more clearly when it was exposed to antibiotics in the A-O wastewater treatment systems on day 1, 100, 200, 240, 330, and 420 (Appendix A). Network analysis was employed to explore the co-occurrence between major microbial genera and the corresponding ARGs in control system A (no antibiotic exposure) and in system B (exposed to TC and SMX in combination). The comparisons were based on significant correlations (Spearman’s r > 0.6, *p* < 0.05). The top 50 genera in each sample were selected for comparative analysis with TC and sulfonamide resistance genes, as well as the *intI1* gene. As shown in Figure 4a, in the absence of antibiotics (system A), the co-occurrence network was divided into three modules with different genera or ARGs. Under the pressure of combined antibiotics in system B, the co-occurrence associations between different microbials and their corresponding ARGs seemed to be more concentrated than that in system A (Figure 4b), which indicated that the evolution of ARGs was strongly impacted by the bacteria community under antibiotic pressure.

In system A, the most abundant tetracycline and sulfonamide resistance genes (*tetC* and *sul1*), as well as the only enzymatic inactivation gene (*tetX*) were mainly associated with about 25 genera (the purple module in Figure 4a). These genera included *Ferruginibacter* (3.09%), *Sediminibacterium* (2.80%), *Candidatus Accumulibacter* (2.74%), *Haliscomenobacter* (2.67%), *Rhodobacter* (1.94%), and *Runella* (1.56%), etc. of which, 14 of the 25 genera were also closely related with *tetA*, *tetO*, *tetW*, *sul1,* and *sul2*. These results suggested that bacterial genera already carried more than one antibiotic resistance gene. Another module included 15 possible host bacterial genera (the red module in Figure 4a) that exhibited an association with *intI1* and *tetL*. This group included *Dechloromonas* (2.29%), *Phaeodactylibacter* (1.17%), *Aeromonas* (0.62%), *Rubellimicrobium* (0.61%), *Streptococcus* (0.48%), and *Hydrogenophaga* (0.44%) (Appendix A).

As shown in Figure 3 and Appendix A, the addition of antibiotics (especially TC and SMX in combination) in system B was of great benefit to the propagation or transmission of ARGs and obviously changed the composition of the microbial community structure compared to those in system A. Exposure of microorganisms to antibiotics was expected to selectively change the relationships between microbes and ARGs. Compared to system A, the distribution of various ARGs in the microbial community was more concentrated in some bacterial genera than others, as shown in Figure 4b. The core bacteria in system B were separated into three groups. The largest of these included more than 30 potential bacterial hosts (the purple module in Figure 4b). Included in this group were *Dechloromonas* (11.58%), *Candidatus Accumulibacter* (4.99%), *Aeromonas* (2.41%), *Rubrivivax* (1.07%), *Roseomonas* (1.06%), and *Rhodobacter* (0.36%), etc. of which 20 were simultaneously related with all the detected tetracycline resistance genes (*tetA*, *tetC*, *tetL*, *tetO*, *tetW,* and *tetX*) and the most abundant sulfonamide resistance gene (*sul1*). Nine genera in particular (*Candidatus Accumulibacter*, *Aeromonas*, *Haliscomenobacter*, *Rheinheimera*, *Rubrivivax*, *Hydrogenophaga*, *Terrimonas*, *Azospira,* and *Acetoanaerobium*) were also found to have relationships with *sul2* and *int1* under antibiotics pressure. These bacteria possibly carried multiple antibiotic resistance and played major roles in the spread of ARGs. Compared to the 14 possible hosts carrying multiple ARGs in system A, only five genera (*Chitinophaga*, *Ferribacterium*, *Ferruginibacter*, *Nitrosomonas,* and *Pseudomonas*) appeared in both systems A and B. The other 15 genera most likely were newly developed genera and possibly carried multiple ARGs under the pressure of TC and SMX, such as *Candidatus Accumulibacter*, *Rubrivivax*, *Rheinheimera*, *Haliscomenobacter,* and *Runella*, etc.

The 20 potential bacterial hosts that showed an intimate relationship with almost all the detected ARGs might possess capabilities to protect themselves against the toxicity of antibiotics and even to degrade antibiotics. Additional research should be continued to study these core genera as they probably carry multiple resistance genes and may evolve into multiple antibiotic resistant bacteria, especially under antibiotics pressure. Previously, *Dechloromonas*, *Candidatus Accumulibacter,* and *Rhodobacter* were reported as the global core bacterial community in 1200 samples taken from 269 wastewater treatment plants in more than 20 countries [51]. *Dechloromonas* and *Candidatus Accumulibacter* were also reported to be related with *tetA* and *sul1* under the pressure of tetracycline and sulfonamide [16]. Wang et al. (2019) reported that the tolerance of these two genera was probably promoted though acquiring TC resistance genes from other microbes under selective TC pressure [46]. Furthermore, *Rhodobacter* has been known for its excellent adaptability to different living environment and this genus was reported to contain the *sul1* gene [52,53]. In this study, the addition of TC and SMX to system B mainly increased the abundance of *Dechloromonas* (13.47-fold) and *Candidatus Accumulibacter* (4.80-fold) compared to their abundance in system A (Appendix A), and these changes closely matched the increasing abundance of ARGs (Figure 3).

The second group included 12 genera (the green module in Figure 4b) that correlated only with genes *sul2* and *int1*. These genera should be related to SMX exposure and in fact, their abundance was obviously increased under SMX pressure. Compared to their abundance in system A, the abundance of *Thiothrix* (9.90% increase), *Candidatus Accumulibacter* (4.99%), *Ferruginibacter* (2.41%), *Pseudomonas* (2.36%), *Flavobacterium* (2.25%), *Runella* (1.94%), and *Thauera* (0.73%) all increased. Moreover, the *sul3* gene was correlated with *Dechloromonas* (11.58%), *Roseomonas* (1.06%), *Sphaerotilus* (0.93%), *Sphingopyxis* (0.49%), *Filimonas* (0.37%), and *Rhodobacter* (0.36%) (Figure 4a and Appendix A).

The *intI1* integron is regarded as the most important and sensitive indicator of bacteria’s ability to incorporate exogenous gene cassettes [15]. Most genera associated with *intI1* also showed a close relationship with other detected ARGs, except for *sul3*. This indicated that *intI1* was an important component in the widespread of ARGs. The co-occurrence associations among these ARGs (especially *intI1*) and their potential bacterial hosts may result in the propagation and transmission of antibiotic resistance through vertical and horizontal gene transfer under antibiotics pressure, thereby facilitating the transfer of antibiotics resistance more widely among bacteria.

*Thiothrix* and *Sphaerotilus* are both well-known filamentous bacteria that are involved in sludge bulking, and have been validated as the main bulking causative bacteria in the type of A-O wastewater treatment system examined in this study [31,54,55]. The abundance of the *sul2* gene showed an obvious tendency to increase when exposed to antibiotics in system B, which was possibly related to the higher percentage of *Thiothrix* that resulted from the exposure of microorganisms to TC and SMX (Figure 3 and Appendix A). Previous research also showed that the abundance of *Thiothrix* was increased when exposed to 50 μg·L^−1^ SMX in a laboratory-scale sequencing batch reactor for two months [24]. The overgrowth of filamentous bacteria might make other bacteria susceptible to being washed out from the system and might be associated with sludge bulking. Therefore, both sludge bulking and bacterial washout will inevitably impact bacterial groups harboring different ARGs and further change the inducement and transmission of ARGs.

*Thauera* and *Pseudomonas*, both of which carry more than three ARGs, have a versatile capacity for antibiotic degradation; similarly, *Ferruginibacter* is known as a type of floc-forming bacterium and is dominant under the pressure of SMX [37,56,57]. However, in the current study, exposure to TC and SMX in combination resulted in the absence of *Ferruginibacter* and the inhibition of *Thauera* (0.84%) and *Pseudomonas* (0.95%) on day 420 in system B (Appendix A). Other researchers have also confirmed that *Thauera*, *Pseudomonas,* and *Ferruginibacter* decreased in relative abundance under different antibiotic pressures (TC, SMX, ofloxacin, or ampicillin), indicating that these genera were not able to endure the biotoxicity of the four antibiotics [1,26,27,46].

The relationships between microbes and ARGs under antibiotic pressure were obviously shown by the network analysis. However, these relationships must be further validated using other approaches, such as culture-dependent methods, metagenomics sequencing, and gene chip technology [18,58]. Furthermore, it must be emphasized that more attention should be devoted to the combined pollution of antibiotics and their metabolic products because antibiotics usually do not exist separately in natural environment or in wastewater treatment plants. The co-existence of antibiotics will inevitably have influence on the microbes and ARGs will especially be harmful to certain microbial communities. Therefore, the impacts of various co-existing antibiotics and their metabolic products on the microbial communities and ARGs should be investigated further.

## 4. Conclusions

In conclusion, antibiotics such as TC and SMX could be partially removed or degraded in the A-O wastewater treatment system. Exposure of sub-inhibition concentrations of antibiotics, especially combinations of different antibiotics, could substantially increase the corresponding ARGs. However, with the antibiotic exposure, the ARGs seemed to be more concentrated in some specific core bacterial genera resulting in production and evolution of multiple antibiotic resistant bacteria, in which the *intI1* had extensive relationships with these bacterial genera. Although further research using other approaches (such as culture-dependent methods) was required to validate the relationships between ARGs and specific bacterial groups, this is the first report about the evolution relationship between antibiotic resistance and microbial community composition in a wastewater treatment system under long-term exposure of sub-inhibition concentrations of combined antibiotics. Considering the antibiotic contaminant facts in most wastewater treatment plants, the influence of lower concentrations of antibiotics exposure, more types of antibiotic combinations, or even the antibiotic metabolites on the evolution of antibiotic resistance genes should be further investigated and clarified in the future research.

## Figures and Tables

**Figure 1 ijerph-16-04681-f001:**
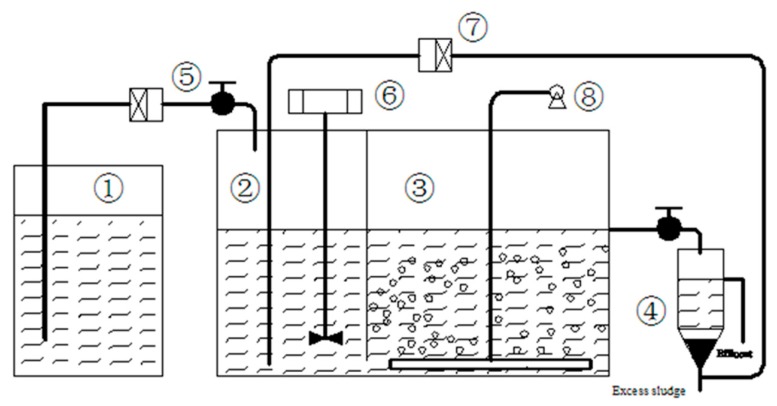
Schematic diagram of the anoxic-aerobic system (① feed tank; ② anoxic zone; ③ aerobic zone; ④ settling tank; ⑤ influent pump; ⑥ stirrer; ⑦ recycle pump; ⑧ air pump).

**Figure 2 ijerph-16-04681-f002:**
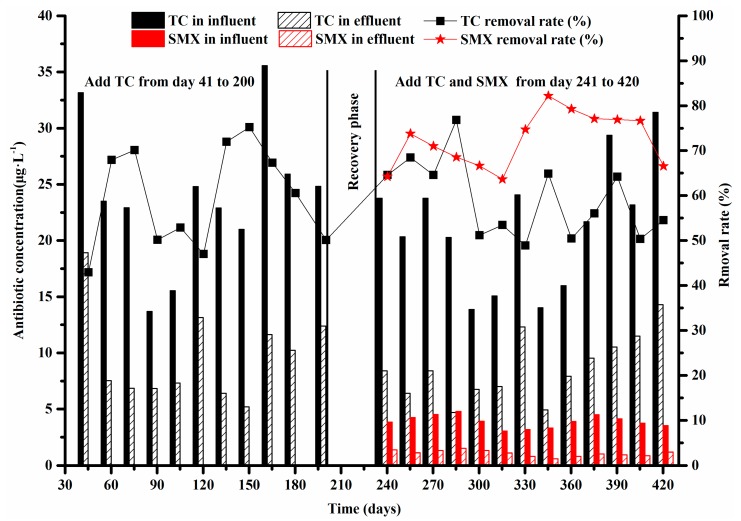
The removal rate (%) and concentrations (μg·L^−1^) of tetracycline (TC) and sulfamethoxazole (SMX) in system B.

**Figure 3 ijerph-16-04681-f003:**
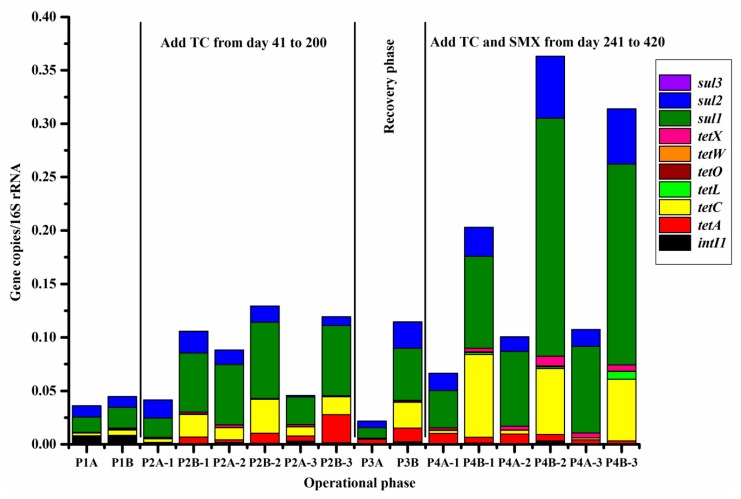
The relative abundances (gene copies of antibiotic resistance genes (ARGs) normalized to the gene copies of 16S rRNA) of six tetracycline (TC) resistance genes (*tetA*, *tetC*, *tetL*, *tetO*, *tetW,* and *tetX*) and three sulfonamide resistance genes (*sul1*, *sul2,* and *sul3*), as well as the integrase gene of class 1 integrons (*intI1*) in different phases in system A and system B.

**Figure 4 ijerph-16-04681-f004:**
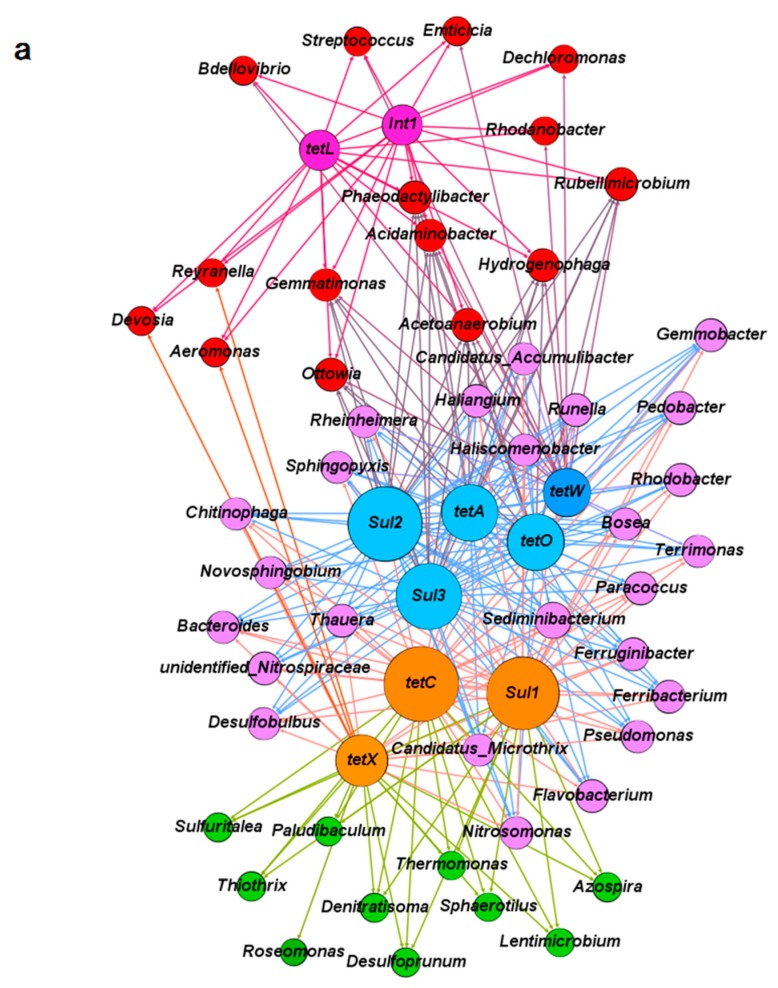
Network analysis between the microbial genera and corresponding antibiotic resistance genes in the anoxic-aerobic wastewater treatment systems based on Spearman’s rank correlations: (**a**) In control system A; (**b**) in system B exposed to antibiotics. Different colors represent the different weighting degrees of nodes and edges. The size of each node is proportional to the number of connections.

**Table 1 ijerph-16-04681-t001:** Primer sets used for quantitative polymerase chain reaction analysis.

Target Gene	Primer Name	Primer Sequence (5′-3′)	Amplicon Size (bp)	Annealing Temperature (°C)	Reference
16Sr RNA	338-F	ACTCCTACGGGAGGCAGCAG	181	55	[13]
	518-R	ATTACCGCGGCTGCTGG			
*tetA*	*tetA*-F	GCTACATCCTGCTTGCCTTC	210	57	[9]
	*tetA*-R	CATAGATCGCCGTGAAGAGG			
*tetC*	*tetC*-F	CTTGAGAGCCTTCAACCCAG	418	60	[13]
	*tetC*-R	ATGGTCGTCATCTACCTGCC			
*tetL*	*tetM*-F	AGTGGAGAAATCCCTGCTCGGT	149	60	[9]
	*tetM*-R	TGACTATTTGGACGACGGGGCT			
*tetO*	*tetO*-F	ACGGARAGTTTATTGTATACC	171	55	[11]
	*tetO*-R	TGGCGTATCTATAATGTTGAC			
*tetW*	*tetW*-F	GAGAGCCTGCTATATGCCAGC	168	60	[11]
	*tetW*-R	GGGCGTATCCACAATGTTAAC			
*tetX*	*tetX*-F	AGCCTTACCAATGGGTGTAAA	278	60	[13]
	*tetX*-R	TTCTTACCTTGGACATCCCG			
*sul1*	*sul1*-F	CGCACCGGAAACATCGCTGCAC	163	60	[40]
	*sul1*-R	TGAAGTTCCGCCGCAAGGCTCG			
*sul2*	*sul2*-F	CTCCGATGGAGGCCGGTAT	190	60	[2]
	*sul2*-R	GGGAATGCCATCTGCCTTGA			
*sul3*	*sul3*-F	TCCGTTCAGCGAATTGGTGCAG	128	60	[40]
	*sul3*-R	TTCGTTCACGCCTTACACCAGC			
*intI1*	*intI1*-F	CCTCCCGCACGATGATC	280	60	[35]
	*intI1*-R	TCCACGCATCGTCAGGC			

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
