# Peer review of "Evolution of Antibiotic Resistance and the Relationship between the Antibiotic Resistance Genes and Microbial Compositions under Long-Term Exposure to Tetracycline and Sulfamethoxazole"

_ijerph, 2019, doi:10.3390/ijerph16234681_

Round 1

Reviewer 1 Report

 This manuscript describes the evolution of antibiotic resistance in an anoxic-aerobic system. Article is good and well presented. It is useful for more researchers. I strongly recommend the publication of this paper.

- There are several typos and grammar issues that must be corrected.

- The main advantages and disadvantages of present study must be added.

- rationale behind choosing TC and SMX

- Authors should justify the limitations of the study if any.

- need more information on qPCR analysis, conditions,       
-   Conclusions are quite obscured and suffer from the lack of specific results. The conclusion should be rewritten.
- It is needed to improve the literature review.

- Figure and table captions should be controlled.

- Please ensure high quality of the figures.

Author Response

Response to Reviewers

Dear Editors and Reviewers:

Thank you for your comments on our submission entitled “Evolution of antibiotic resistance and the relationship between the antibiotic resistance genes and microbial compositions under long time exposure to tetracycline and sulfamethoxazole” (ID: ijerph-633251). We are thankful for your letter and for the reviewers’ comments concerning our manuscript. We have responded to all these comments and also revised the manuscript accordingly. Revisions to the manuscript are indicated in red fronts in the revised version. All authors have reviewed the manuscript and approved the resubmission of the manuscript. Thank you very much for your attention and consideration.

Response to Reviewer comment No. 1

 This manuscript describes the evolution of antibiotic resistance in an anoxic-aerobic system. Article is good and well presented. It is useful for more researchers. I strongly recommend the publication of this paper.
- There are several typos and grammar issues that must be corrected.

Response: Thank you for your comments. We have corrected these typos and grammar issues carefully and listed the revsied description here point by point. The revised parts are marked in red fronts in the revised version.
Details:

Comment: 1. - The main advantages and disadvantages of present study must be added.

Response: Thank you for your suggestion. The main advantages and disadvantages of present study have been added in the Conclusions part line 418-426 as indicated in red fonts.

Comment: 2.- rationale behind choosing TC and SMX

Response: Thank you for your suggestion. The reason for choosing TC and SMX has been added in the Introduction part line 70-77 as indicated in red fonts.

Comment: 3.- Authors should justify the limitations of the study if any.

Response: Thank you for your suggestion. The limitation of this study is that there may be lower concentrations of antibiotics, or more types of antibiotic combinations, in the actual sewage treatment system. So, it is necessary to study the situation of lower antibiotic concentration exposure and more kinds of antibiotics exposure in the future research. We have showed these details in the Conclusions part line 423-426 in red fonts.

Comment: 4.- need more information on qPCR analysis, conditions,       
Response: Thank you for your suggestion. The main conditions for qPCR analysis was added in the Materials and method section 2.4 line 151-155 as indicated in red fonts.

Comment: 5.- Conclusions are quite obscured and suffer from the lack of specific results. The conclusion should be rewritten.
Response: Thank you for your suggestion. The conclusion has been be rewritten and the main results of our study was contained.

Comment: 6.- It is needed to improve the literature review.
Response: Thank you for your suggestion. We have improved the literature review and rearrange the number of corresponding references.

Comment: 7.- Figure and table captions should be controlled.
Response: Thank you for your suggestion. The captions of figures and tables have been re-structured.
Comment: 8.- Please ensure high quality of the figures.

Response: Thank you for your suggestion. The quality of the figures has been improved and the resolution was more than 300 dpi.

Thanks again for your comments and suggestions. We hope it can be up to your requirements this time. However, if you have any other question, please feel free to contact me.

Best wishes,

Prof. Qingxiang Yang

College of Life Sciences, Henan Normal University

    46 Jianshe Road, Xinxiang 453007, Henan Province, China.

    Email: [email protected]; [email protected].

Reviewer 2 Report

In this paper the authors described the results of an experimental scale wastewater system utilized to study the evolution of antibiotic resistance and of the fate of the ARGs by long-term exposure, of two chosen antibiotic, both in singular than in combined mixture (tetracycline and tetracycline plus sulfamethoxazole) .

The results are very interesting and innovative, underlining the importance of deeply study this topic especially to better understand the spreading of antibiotic resistance in the environment and the role of wastewater treatment plant to reduce this effect or otherwise to favorite the spreading.

In my opinion the manuscript is clear, well written and could be published in the journal.

Author Response

Response to Reviewers

Dear Editors and Reviewers:

Thank you for your comments on our submission entitled “Evolution of antibiotic resistance and the relationship between the antibiotic resistance genes and microbial compositions under long time exposure to tetracycline and sulfamethoxazole” (ID: ijerph-633251). We are thankful for your letter and for the reviewers’ comments concerning our manuscript. We have responded to all these comments and also revised the manuscript accordingly. Revisions to the manuscript are indicated in red fronts in the revised version. All authors have reviewed the manuscript and approved the resubmission of the manuscript. Thank you very much for your attention and consideration.

Response to Reviewer comment No. 2

In this paper the authors described the results of an experimental scale wastewater system utilized to study the evolution of antibiotic resistance and of the fate of the ARGs by long-term exposure, of two chosen antibiotic, both in singular than in combined mixture (tetracycline and tetracycline plus sulfamethoxazole) .

The results are very interesting and innovative, underlining the importance of deeply study this topic especially to better understand the spreading of antibiotic resistance in the environment and the role of wastewater treatment plant to reduce this effect or otherwise to favorite the spreading.

In my opinion the manuscript is clear, well written and could be published in the journal.

Response:

We are grateful for this positive comment on our manuscript. We have checked the manuscript again carefully and corrected some mistakes.

Thanks again for your comments and suggestions. We hope it can be up to your requirements this time. However, if you have any other question, please feel free to contact me.

Best wishes,

Prof. Qingxiang Yang

College of Life Sciences, Henan Normal University

    46 Jianshe Road, Xinxiang 453007, Henan Province, China.

    Email: [email protected]; [email protected].

Reviewer 3 Report

The manuscript examines the evolution of antibiotic resistance and the relationship between ARGs and microbial communities under long-term antibiotic (TC and SMX) selection pressure were investigated in a laboratory-scale A-O wastewater treatment system. The experiments are well thought out and appropriate controls are present. While this is an interesting paper and is worthy of publishing there are some minor mistakes.
For instance, removal instead of removals in the abstract. The manuscript needs to be further proof-read. In addition, some mistakes are pointed out as described below:
Line 45 residual antibiotics of antibiotic residue.
In line 46, author confuses misuse with the sole existence of the antibiotics.
In line 51 antibiotic contaminants instead of antibiotics.
Sentence is unclear and should be rephrased in line 83.
Line 69 correct spelling finstead or “adiing” and delete Then,.
Line 156-161, this would be better placed in the materials and method section instead of result and discussion.
Line 412, delete in the.
However, these mistakes do not detract the reader from the scientific meaning therefore, the publication of the current manuscript is recommended after correction.

Author Response

Response to Reviewers

Dear Editors and Reviewers:

Thank you for your comments on our submission entitled “Evolution of antibiotic resistance and the relationship between the antibiotic resistance genes and microbial compositions under long time exposure to tetracycline and sulfamethoxazole” (ID: ijerph-633251). We are thankful for your letter and for the reviewers’ comments concerning our manuscript. We have responded to all these comments and also revised the manuscript accordingly. Revisions to the manuscript are indicated in red fronts in the revised version. All authors have reviewed the manuscript and approved the resubmission of the manuscript. Thank you very much for your attention and consideration.

Response to Reviewer comment No. 3

The manuscript examines the evolution of antibiotic resistance and the relationship between ARGs and microbial communities under long-term antibiotic (TC and SMX) selection pressure were investigated in a laboratory-scale A-O wastewater treatment system. The experiments are well thought out and appropriate controls are present. While this is an interesting paper and is worthy of publishing there are some minor mistakes. For instance, removal instead of removals in the abstract. The manuscript needs to be further proof-read. In addition, some mistakes are pointed out as described below:

However, these mistakes do not detract the reader from the scientific meaning therefore, the publication of the current manuscript is recommended after correction.
Response: Thank you for your recognition of our study. We have checked the manuscript again carefully and addressed all comments raised by the reviewer accordingly in red fronts in the revised version.

Comment: 1. Line 45 residual antibiotics of antibiotic residue.

Response: Thank you for your kind suggestion. We have replaced “antibiotic residue” with “residual antibiotics” in the revised version.
Comment: 2. In line 46, author confuses misuse with the sole existence of the antibiotics.
Response: Thank you for your suggestion. This sentence has been rewritten as “There is an indisputable fact that the abuse of antibiotics can lead to proliferation, accumulation and widespread distribution of antibiotic resistance bacteria and antibiotic resistance genes (ARGs), which is becoming an increasing threat to global health and posing a serious threat to the environment” in the revised version.

Comment: 3. In line 51 antibiotic contaminants instead of antibiotics.
Response: Thank you for your suggestion. We have replaced “antibiotics” with “antibiotic contaminants” in the revised version.

Comment: 4. Sentence is unclear and should be rephrased in line 83.
Response: Thank you for your suggestion. This sentence has been rephrased as “However, it was not clear about the evolution of antibiotic resistance and the relationship between ARGs and microbial community compositions in activated sludge, especially under long-term antibiotic pressure. Tracking the shifts of key microbial genera and further investigating the succession of the potential ARGs hosts will help us to understand the evolution process and mechanism of ARGs transmission under antibiotic pressure in wastewater treatment system”.

Comment: 5. Line 69 correct spelling finstead or “adiing” and delete Then,.
Response: Thank you for your kind suggestion. We are sorry for our errors and have corrected it. Also, the redundant word “Then” has been deleted.
Comment: 6. Line 156-161, this would be better placed in the materials and method section instead of result and discussion.
Response: Thank you for your suggestion. We have moved this description into the Materials and method section 2.1 line 110-113 as indicated in red fonts.

Comment: 7. Line 412, delete in the.
Response: Thank you for your kind suggestion. We have deleted it.

Thanks again for your comments and suggestions. We hope it can be up to your requirements this time. However, if you have any other question, please feel free to contact me.

Best wishes,

Prof. Qingxiang Yang

College of Life Sciences, Henan Normal University

    46 Jianshe Road, Xinxiang 453007, Henan Province, China.

    Email: [email protected]; [email protected].
